# ROBUST INVERSE REINFORCEMENT LEARNING UNDER STATE ADVERSARIAL PERTURBATIONS

## ABSTRACT

State adversarial perturbations –such as sensor noise, environmental interference, or targeted attacks– are common in real-world systems, often leading to compromised state observations. Despite this, Inverse Reinforcement Learning (IRL) in the context of State-Adversarial Markov Decision Processes (SA-MDPs) has received limited attention, primarily because conventional notions of optimality do not apply. In this paper, we introduce a novel definition of optimality that ensures the existence of an optimal policy within SA-MDPs. Building on this foundation, we propose the State-Adversarial Max-Margin IRL (SAMM-IRL) algorithm, designed to be robust against state perturbations. Our theoretical analysis, supported by empirical validation, demonstrates that SAMM-IRL significantly enhances IRL performance in adversarial environments, providing a robust framework for real-world applications that demand resilience.

## 1 INTRODUCTION

Many real-world systems, such as autonomous vehicles and robotics, are vulnerable to disruptions in their sensory data, whether caused by environmental factors such as noise or deliberate adversarial attacks. These state adversarial perturbations distort the agent's perception of the environment, resulting in compromised decision-making. In such settings, standard learning approaches, including Inverse Reinforcement Learning (IRL), struggle to maintain performance because they rely on the assumption of accurate state observations. This challenge is particularly critical in safety-sensitive applications, where even small errors in state estimation can lead to significant risks.

Traditional IRL methods have been successful in recovering reward functions and replicating expert behavior. While some state-of-the-art approaches can handle limited static noise, they are not equipped to deal with dynamic adversarial perturbations that actively interfere with state observations. This makes these methods vulnerable when such assumptions are violated by adversarial forces. Our work focuses on State-Adversarial Markov Decision Processes (SA-MDPs) as formalized by Zhang et al. (2020), where adversaries perturb state observations. This setting presents a challenge, as a policy that is optimal for one initial state may become suboptimal for another. To address this, we introduce Resilient State Optimality (RSO), which redefines optimality by maximizing the expected value across initial state distributions under adversarial conditions. This guarantees the existence of robust policies, even under worst-case adversarial perturbations, enabling more resilient policy recovery.

Building on this foundation, we propose State-Adversarial Max-Margin IRL (SAMM-IRL), a novel algorithm that extends the Max-Margin IRL framework (Abbeel & Ng, 2004) to address the challenge of robust IRL under adversarial environments. We provide theoretical performance guarantees for policies learned by SAMM-IRL, and our empirical results demonstrate effectiveness in adversarially perturbed environments.

## 2 RELATED WORK

IRL has been extensively studied as a method to infer reward functions from expert demonstrations, allowing agents to learn in environments without explicit rewards. Ng & Russell (2000) laid the foundation for IRL by recovering reward functions from observed expert behavior. Max-Margin

IRL (Abbeel & Ng, 2004) improved this by focusing on feature expectation matching, addressing under-determination in reward recovery. However, these methods assume stable environments with accurate observations, making them vulnerable to adversarial attacks on state observations.

Several approaches have been proposed to address adversarial challenges in Reinforcement Learning (RL). Robust Adversarial Reinforcement Learning (RARL) (Pinto et al., 2017) modeled the interaction between agent and adversary as a two-player Markov game, where the adversary can modify environment parameters. Zhang et al. (2020) formalized SA-MDPs and demonstrated that an optimal policy may not exist under dynamic adversarial perturbations, proposing methods to improve robustness against a fixed adversary. Their follow-up work (Zhang et al., 2021) explored learned adversaries actively perturbing state observations during training, improving robustness despite lacking convergence guarantees.

Recent IRL methods, such as Generative Adversarial Imitation Learning (GAIL) (Ho & Ermon, 2016) and Adversarial Inverse Reinforcement Learning (AIRL) (Fu et al., 2018), use adversarial training frameworks inspired by Generative Adversarial Networks (GAN) to enhance policy imitation and reward recovery. However, both GAIL and AIRL assume assume relatively accurate state observations, leaving them vulnerable to adversarial manipulations that target the agent's perception of the environment.

In contrast, our work leverage RSO as a framework that guarantees the existence of optimal policies in SA-MDPs, enabling the development of both RL and IRL methods that can withstand adversarial manipulation of state observations. Building on this, we propose SAMM-IRL, which adapts Max-Margin IRL to the SA-MDP framework, providing theoretical guarantees for robust policy and reward recovery under adversarial conditions. Empirical results demonstrate significant improvements in adversarial environments, positioning SAMM-IRL as a robust alternative to existing methods.

## 3 BACKGROUND

In the following, we first briefly review the MDP framework, followed by the traditional IRL problem, and then review the Max-Margin IRL algorithm from Abbeel & Ng (2004) to solve the aforementioned IRL problem for MDPs. This will provide insights on how to address the IRL problem under state adversarial perturbations.

### 3.1 MARKOV DECISION PROCESS

An MDP $\mathcal{M}$ is defined by a tuple $(\mathcal{S}, \mathcal{A}, R, P, P_0, \gamma)$ where $\mathcal{S}$ the set of states, $\mathcal{A}$ the set of actions, $R$ the reward function, $P$ the state transition distribution, $P_0$ the initial state distribution, and $\gamma$ the discount factor. Given an initial state $S_0 = s$, the performance of a policy $\pi$ is measured by the expected discounted return, defined by the state-value function $V_\pi(s)$.

$$V_\pi(s) = \mathop{\mathbb{E}}_{\substack{S_{t+1} \sim P(\cdot|S_t, A_t) \\ A_t \sim \pi(\cdot|S_t)}} \left[ \sum_{t=0}^{\infty} \gamma^t R(S_t, A_t, S_{t+1}) \,\middle|\, S_0 = s \right]. \tag{1}$$

The goal is to find a policy that maximizes the state-value function in every state. Formally, $\pi^*$ is an optimal policy for a given MDP if and only if

$$V_{\pi^*}(s) \geq V_\pi(s), \quad \forall s \in \mathcal{S}, \quad \forall \pi. \tag{2}$$

### 3.2 INVERSE REINFORCEMENT LEARNING

The seminal IRL problem includes an MDP without a reward function, i.e., $\mathcal{M}/R$, and an expert policy $\pi_E$ demonstrated through trajectories $\mathcal{D} = \{\tau_1, \ldots, \tau_m\}$, where each $\tau_i$ includes state-action pairs $(s_t, a_t)$. The challenge is to infer the unknown reward function $R^{\mathbf{w}}$, parameterized by an unknown reward weight vector $\mathbf{w} \in \mathbb{R}^k$, from $\mathcal{D}$.

For tractability, we use linear reward functions, i.e., $R^{\mathbf{w}}(s, a, s') = \langle \mathbf{w}, \phi(s, a, s') \rangle$, where $\phi \in \mathbb{R}^k$ represents the features of the state-action pair: $\phi : \mathcal{S} \times \mathcal{A} \times \mathcal{S} \to [0, 1]^k$. For state-only rewards, the feature function becomes $\phi : \mathcal{S} \to [0, 1]^k$. The feature expectation under a policy $\pi$ is defined

as $\mu_\pi(s) = \mathbb{E}_\pi \left[ \sum_{t=0}^{\infty} \gamma^t \phi(S_t, A_t, S_{t+1}) \middle| S_0 = s \right]$. Thus, the expected discounted return, or the state-value function $V_\pi(s)$, can be expressed as

$$V_\pi^{\mathbf{w}}(s) = \mathbb{E}_{\substack{S_{t+1} \sim P(\cdot|S_t, A_t) \\ A_t \sim \pi(\cdot|S_t)}} \left[ \sum_{t=0}^{\infty} \gamma^t R^{\mathbf{w}}(S_t, A_t, S_{t+1}) \middle| S_0 = s \right] = \langle \mathbf{w}, \mu_\pi(s) \rangle. \quad (3)$$

The goal of the IRL problem is to find the reward weights $\mathbf{w}$ that maximize the Eq. 3, i.e., $\mathbf{w}^* = \arg\max_{\mathbf{w}} V_\pi^{\mathbf{w}}(s)$. Having the optimal weights $\mathbf{w}^*$, we can then find the optimal policy $\pi^*$ that satisfies Equation 2 as $\pi^* = \arg\max_\pi V_\pi^{\mathbf{w}^*}(s)$.

### 3.3 MAX-MARGIN IRL ALGORITHM

The Max-Margin IRL algorithm for MDPs begins by estimating feature expectations for an initial policy. It iteratively finds a reward function that maximizes the margin, the difference between the expert's feature expectations and those from the current and previous policies. This margin adjusts the reward weights and the policy is updated accordingly. This process repeats, refining both the reward function and the policy until the margin is sufficiently small, resulting in a policy that closely approximates the expert's behavior. The procedure is detailed in Algorithm 1.

---

**Algorithm 1** Max-Margin Algorithm for Apprenticeship Learning in MDPs

---

1: **INPUT**: Expert trajectories $\{\tau_E^k\}_{k=1}^m$
2: Calculate expert's feature expectations $\mu_E$
3: Initialize: Randomly pick an initial policy $\pi^{(0)}$
4: Calculate the feature expectations $\mu^{(0)}$ of the initial policy
5: Set iteration counter $i = 1$
6: **repeat**
7:     Compute $\mathbf{w}^{(i)} = \underset{\mathbf{w}: \|\mathbf{w}\|_2 \leq 1}{\arg\max} \ \underset{j \in \{0, \dots, i-1\}}{\min} \mathbf{w}^T (\mu_E - \mu^{(j)})$
8:     Compute the optimal $\pi^{(i)}$ with $R = \mathbf{w}^{(i)T} \phi$ using Equation 2
9:     Generate new trajectories and calculate $\mu^{(i)}$
10:    Increment iteration counter $i = i + 1$
11: **until** $\|\mu_E - \mu^{(i)}\|_2 \leq \varepsilon$
12: **OUTPUT**: Last policy $\pi^{(i_\varepsilon)}$ and weight $\mathbf{w}^{(i_\varepsilon)}$

---

In step 7 of Algorithm 1, we compute the unit normal vector $\mathbf{w}^{(i)}$ defining the hyperplane that maximally separates $\mu_E$ from $\mu^{(0)}, \dots, \mu^{(i-1)}$. This involves solving a min-max optimization problem to find a vector $\mathbf{w}^{(i)}$, which maximizes the minimum distance between the feature expectations of the expert's policy $\mu_E$ and the feature expectations of other policies $\{\mu^{(j)}\}_{j=0}^{i-1}$. The key assumption is that the expert policy $\pi_E$ is *optimal* under condition (2), implying that for weight vector $\mathbf{w}^{(i)}$, we have $\mathbf{w}^{(i)T} \cdot \mu_E > \mathbf{w}^{(i)T} \cdot \mu_\pi$ for all other policies $\pi$.[1]

## 4 STATE-ADVERSARIAL MDPS AND SAMM-IRL ALGORITHM

An SA-MDP extends the MDP framework by introducing an adversary that perturbs the agent's observations of the environment. Formally, an SA-MDP defined by the tuple $\tilde{\mathcal{M}} = (\mathcal{S}, \mathcal{A}, B, R, P, P_0, \gamma)$, which extends the standard MDP by introducing an additional mapping $B : \mathcal{S} \to 2^{\mathcal{S}}$, where $2^{\mathcal{S}}$ represents the power set of $\mathcal{S}$ (Zhang et al., 2020). An *adversary* $\nu$ maps the actual state $s$ to a perturbed state $\nu(s) \in B(s)$. The perturbation set $B(s)$,[2] restricts the adversary to

---

[1]This inequality arises because the expert's expected return under $\mathbf{w}^{(i)}$ exceeds that of any other policy, placing $\mu_E$ outside the convex hull of the feature expectations $\{\mu^{(j)}\}_{j=0}^{i-1}$ of suboptimal policies. By the separation theorem, a hyperplane with $\mathbf{w}^{(i)}$ as its normal vector separates $\mu_E$ from $\{\mu^{(j)}\}_{j=0}^{i-1}$.

[2]The theoretical justification for this extension is provided in Zhang et al. (2020) through the proof of Lemma 1. The proof demonstrates that given an SA-MDP, $\tilde{\mathcal{M}}$, and a fixed policy $\pi$, there exists an MDP, $\mathcal{M}$, such that the optimal policy of $\mathcal{M}$ is the optimal adversary $\nu^*$ for the SA-MDP given the fixed $\pi$, and $\nu^*$ is restricted to the set $K = \{\nu : \forall s, \exists a \in B(s), \nu(a|s) = 1\}$

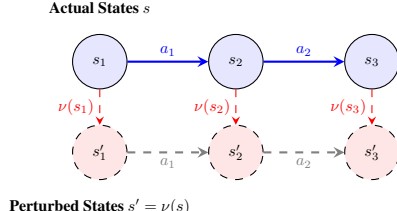

Figure 1: State perturbations in an SA-MDP. Solid blue lines show actions between actual states, dashed red lines indicate perturbations, and gray lines represent actions between perturbed states.

perturb $s$ to a predefined subset of states. This is crucial as it limits perturbations to the predefined set $B(s)$, making the theoretical model robust.

In this framework, the agent's policy $\pi$ becomes a function of the perturbed state, that is, $\pi \circ \nu$ where $\pi \circ \nu = \pi(a|\nu(s))$ and the corresponding value functions are:

$$V_{\pi \circ \nu}(s) = \mathop{\mathbb{E}}_{\substack{S_{t+1} \sim P(\cdot|S_t, A_t) \\ A_t \sim \pi(\cdot|\nu(S_t))}} \left[ \sum_{t=0}^{\infty} \gamma^t R(S_t, A_t, S_{t+1}) \Big| S_0 = s \right], \tag{4}$$

and

$$Q_{\pi \circ \nu}(s, a) = \mathop{\mathbb{E}}_{\substack{S_{t+1} \sim P(\cdot|S_t, A_t) \\ A_t \sim \pi(\cdot|\nu(S_t))}} \left[ \sum_{t=0}^{\infty} \gamma^t R(S_t, A_t, S_{t+1}) \Big| S_0 = s, A_0 = a \right]. \tag{5}$$

**Definition 1** (Optimal Adversary). *Given a policy $\pi$, the* optimal adversary $\nu^*(\pi)$ *is the adversary that minimizes the expected total discounted reward, i.e., $V_{\pi \circ \nu^*}(s) = \min_\nu V_{\pi \circ \nu}(s)$, and $Q_{\pi \circ \nu^*}(s, a) = \min_\nu Q_{\pi \circ \nu}(s, a)$.*

We present the formal theorem from Zhang et al. (2020), confirming that the value functions $V_{\pi \circ \nu^*}$ and $Q_{\pi \circ \nu^*}$ are well-defined for every policy $\pi$.

**Theorem 1** (Existence of the optimal adversary value function (Zhang et al., 2020)). *Let $\pi$ be a policy, and let us define the optimal adversary Bellman operator $\mathcal{T}^\pi : \mathbb{R}^{\mathcal{S}} \to \mathbb{R}^{\mathcal{S}}$ as*

$$(T^\pi V_{\pi \circ \nu})(s) = \min_{\nu(s) \in B(s)} \sum_{a \in \mathcal{A}} \pi(a|\nu(s)) \sum_{s' \in \mathcal{S}} P(s'|s, a) \left[ R(s, a, s') + \gamma V_{\pi \circ \nu}(s') \right]. \tag{6}$$

*The optimal adversary Bellman operator is a contraction w.r.t. the $\| \cdot \|_\infty$, and due to the Banach fixed point theorem, it has a unique fixed point which coincides with $V_{\pi \circ \nu^*}$.*

### 4.1 Resilient State Optimality in SA-MDPs

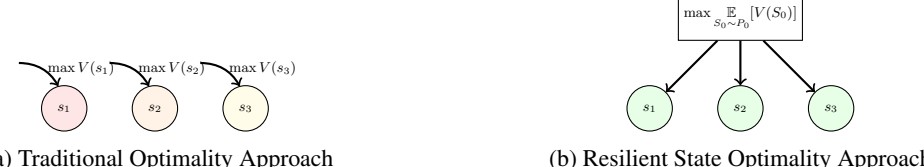

(a) Traditional Optimality Approach        (b) Resilient State Optimality Approach

Figure 2: Comparison of traditional and novel optimality approaches in SA-MDP. Both approaches deal with initial states drawn from $P_0$. The traditional approach tries to maximize each $V(s_i)$ individually and RSO approach maximizes the expected value over the initial state distribution.

In SA-MDPs, conventional optimal policies may not exist due to the dynamic interplay between adversarial perturbations and agent actions as detailed in the following section in Algorithm 2. This algorithm iteratively the policy using a reinforcement learning algorithm, while continuously recalculating the adversary's optimal strategy, ensuring the policy becomes robust against worst-case

adversarial perturbations. In this section, we explore these theoretical challenges and present our approach to addressing them, with further practical details provided in Section 4.2.

To define an optimal policy $\pi^*$ in an SA-MDP under the optimal adversary $\nu^*$, one might simply aim for straight-forward extension of the optimality in MDPs as

$$V_{\pi^* \circ \nu^*(\pi^*)}(s) \geq V_{\pi \circ \nu^*(\pi)}(s) \qquad \forall s \in \mathcal{S}, \forall \pi. \tag{7}$$

Zhang et al. (2020) demonstrated through a three-state, two-action example that an optimal policy may not always exist. The optimality condition in Equation 7 is similar to that in standard MDPs, where the policy must be optimal from any starting state $s$. However, in SA-MDPs, adversarial perturbations make this requirement problematic. To address this challenge, we introduce Resilient State Optimality (RSO), which allows the policy to balance performance across the distribution of initial states rather than being optimal in each state. The following theorem formalizes this new notion, namely Resilient State Optimality (RSO) and proves the existence of an optimal policy under this new condition.

**Theorem 2** (Existence of Optimal Policy under RSO in SA-MDPs). *Let $\tilde{\mathcal{M}}$ be an SA-MDP with a finite state space $\mathcal{S}$ and a finite action space $\mathcal{A}$. Besides, $S_0 \sim P_0$ denotes the initial state and $V_{\pi \circ \nu^\star}$ is the value function of a policy $\pi$ under an optimal adversarial perturbation $\nu^*(\pi)$. Then, there exists a policy $\pi^*_{\text{RSO}} = \pi^* \circ \nu^\star(\pi^*)$ such that*

$$\mathbb{E}_{S_0 \sim P_0}[V_{\pi^*_{\text{RSO}}}(S_0)] \geq \mathbb{E}_{S_0 \sim P_0}[V_{\pi \circ \nu^\star(\pi)}(S_0)], \quad \forall \pi. \tag{8}$$

*Proof Sketch.* To prove the existence of an optimal policy under RSO in SA-MDPs, we leverage the policy space's mathematical properties. With finite states and actions, the set of all possible policies forms a convex and compact space, which means that it is a well-behaved, finite-dimensional set. We show that the function mapping any policy to its expected performance under the worst-case adversarial perturbation is concave and continuous. This implies that there exist a policy $\pi^*_{\text{RSO}}$ maximizing expected performance across all initial states. The complete proof is provided in Appendix A.1.1. ∎

**Remark 1.** *Our theoretical framework naturally extends to Q-value functions, maintaining the same optimality conditions in Q-value-based learning settings. Moreover, the results hold when the initial-state distribution is adversarially perturbed. For the formal statements of these extensions of Theorem 2 and their corresponding proofs, see Appendix A.1.1.*

**Assumption 1.** *We formulate and analyze the IRL problem in an SA-MDP framework using the actual initial-state distribution for theoretical rigor. In the experimental setup, we consider initial states drawn from the perturbed distribution to reflect its practical efficiency.*

## 4.2 Policy Optimization in SA-MDPs

Our theoretical framework based on Equation 8 formulates the worst-case scenario in an SA-MDP as a dynamic interplay between the adversary and the agent. The adversary iteratively updates its policy to minimize the expected return of the agent, while the agent adapts its policy accordingly.

$$\nu^* = \underset{\nu(\cdot) \in B(\cdot)}{\arg\min} \mathbb{E}_{s \sim P_0}\left[V_{\pi^* \circ \nu}(s)\right] \quad \text{s.t.} \quad \pi^* \circ \nu = \underset{\pi}{\arg\max} \mathbb{E}_{s \sim P_0}\left[V_{\pi \circ \nu}(s)\right]. \tag{9}$$

This iterative process, outlined in Algorithm 2, is crucial for optimizing the expert agent's policy and finding the optimal adversarial strategy. We assume that the process stabilizes after enough iterations, allowing the agent to perform robustly under adversarial perturbations. The algorithm aligns with the general structure of reinforcement learning with an integrated adversarial layer.

**Assumption 2.** *We focus primarily on the observer agent's learning process and its convergence within the IRL framework, as outlined in Section 4.3. We address the contraction behavior of the SAMM-IRL algorithm in Theorem 3 (Section 4.3).*

*For the expert agent, we assume convergence after a number of iterations, justified by the existence of an optimal policy in SA-MDPs (Theorem 2). This assumption allows us to concentrate on the observer's learning process. While a detailed convergence analysis for the expert is outside the scope of this work, we briefly summarize the contraction property for the expert's policy optimization using an adaptation of SARSA algorithm (Rummery & Niranjan, 1994) in Section A.4, demonstrating the stability under adversarial perturbations.*

---

**Algorithm 2** Policy Optimization under Non-stationary Optimal Adversarial Perturbations

---

1: **Input**: Initial policy $\pi^{(0)}$
2: Compute initial optimal adversary $\nu^*(\pi^{(0)})$
3: **for** each iteration $i$ until convergence **do**
4:     Update policy $\pi^{(i+1)}$ using RL under $\nu^*(\pi^{(i)})$
5:     Recompute $\nu^*(\pi^{(i+1)})$.
6: **end for**
7: **Output**: Final policy $\pi^*$ and adversary $\nu^*(\pi^*)$

---

### 4.3 SAMM-IRL Algorithm

Building on the notion of RSO, we adapt the feature expectation calculations used in Max-Margin IRL to align with the RSO framework. In conventional Max-Margin IRL, the goal is to recover the expert's reward function by matching feature expectations. However, in the presence of state perturbations, directly applying these methods fails to yield reliable results. The following adaptions of calculating the feature expectations according to RSO allows the algorithm to handle the challenges posed by adversarial perturbations in SA-MDPs.

**Definition 2.** *Let $\pi$ be a policy and $\nu^* = \nu^*(\pi)$ be the corresponding optimal adversary. We define the "actual perturbed feature expectation" of $\pi$ as*

$$\mu_{\pi \circ \nu^*} = \mathbb{E}_{S \sim P_0} \left[ \mathbb{E}_{\pi \circ \nu^*} \left[ \sum_t \gamma^t \phi(S_t) \Big| S_0 = S \right] \right], \tag{10}$$

*where $S_t$ represents the actual states visited.*

**Definition 3.** *We define the "believed perturbed feature expectation" as*

$$\tilde{\mu}_{\pi \circ \nu^*} = \mathbb{E}_{S \sim P_0} \left[ \mathbb{E}_{\pi \circ \nu^*} \left[ \sum_t \gamma^t \phi(\nu^*(S_t)) \Big| S_0 = S \right] \right], \tag{11}$$

*where $\nu^*(S_t)$ represents the believed states.*

The actual perturbed feature expectation $\mu_{\pi \circ \nu^*}$ and the believed perturbed feature expectation $\tilde{\mu}_{\pi \circ \nu^*}$ involve state perturbations in SA-MDPs. $\mu_{\pi \circ \nu^*}$ represents the true states visited by the agent, while $\tilde{\mu}_{\pi \circ \nu^*}$ reflects the states perceived by the agent under perturbations. The key difference is in how the perturbation function $\nu^*$ is applied: $\mu_{\pi \circ \nu^*}$ considers the actual states whereas $\tilde{\mu}_{\pi \circ \nu^*}$ applies $\nu^*$ directly to compute the feature vector. In applications where observation of actual states is infeasible, our algorithm uses $\tilde{\mu}_{\pi \circ \nu^*}$ instead. As shown in Lemma 1, this approximation has an error bounded by $\varepsilon_0$ under bounded perturbations.

We assume the expert agent has an optimal policy, namely, $\langle \mathbf{w}^*, \mu_{\pi_E \circ \nu^*} \rangle \geq \langle \mathbf{w}^*, \mu_{\pi \circ \nu^*} \rangle$, for every policy $\pi$. This condition aligns with the optimality condition in Equation 8, indicating that the expert's policy is optimal under state-adversarial perturbations.

Once the adversary's optimal perturbation policy $\nu^*$ is fixed –following its training with the expert (Figure 3a), which occurs separately from the IRL process– the observer's policy $\pi_{\text{obs}}$ is iteratively updated in response to these fixed perturbations until convergence. The fixed adversary policy serves as input to the SAMM-IRL algorithm, which then focuses on aligning the observer's feature expectations with the expert's under the given adversarial conditions. The key steps are summarized in Algorithm 3 and illustrated in Figure 3b.

In the following we discuss the key properties of the SAMM-IRL algorithm.

1. **Input:** The input to the algorithm consists of trajectories of the expert agent, following the policy returned from the min-max optimization process for expert-adversary training (as described in Algorithm 2) under perturbed conditions.

2. **Optimization in SA-MDPs:** During optimization, the algorithm computes policies that maximize the expected value over the initial state distribution $P_0$ while estimating feature expectations based on perturbed trajectories. This aligns the policy with the new optimality definition for SA-MDPs, ensuring performance in expectation over the initial state distribution.

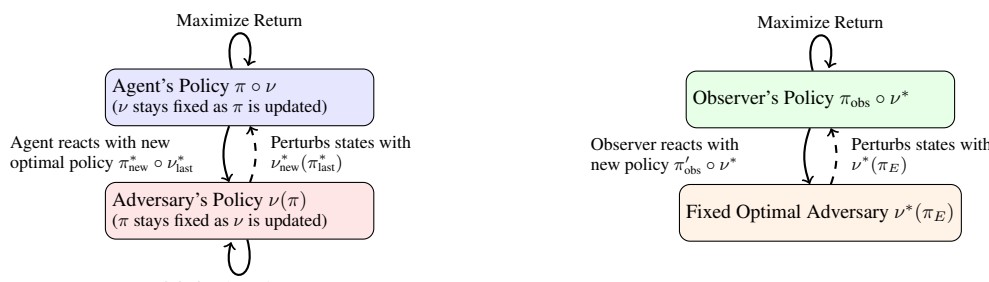

(a) Bi-level optimization process of the expert.          (b) Observer's policy optimization process.

Figure 3: (a) Bi-level optimization process: The agent's policy remains fixed while the adversary optimizes its policy, and vice versa. Each party reacts to the other's optimal policy after one has finished optimizing. (b) The observer updates its policy in response to a fixed optimal adversary $\nu^*(\pi_E)$, with the adversary's strategy unchanged.

---

**Algorithm 3** State-Adversarial Max-Margin IRL in SA-MDPs

---

1: **Input**: Expert trajectories $\{\tau_E^k\}_{k=1}^m$ generated by $\pi_{E\circ\nu^*}$ under state perturbations $\nu^*$, $P_0$
2: Calculate expert's believed perturbed feature expectations $\tilde{\mu}_E$ via Equation 11
3: Initialize: Randomly pick an initial policy $\pi^{(0)}$
4: Compute $\tilde{\mu}^{(0)}$ by Equation 11
5: Set iteration counter $i = 1$
6: **repeat**
7:     Compute $\mathbf{w}^{(i)} = \arg\max_{\mathbf{w}:\|\mathbf{w}\|_2 \leq 1} \min_{j\in\{0,\ldots,i-1\}} \mathbf{w}^T(\tilde{\mu}_E - \tilde{\mu}^{(j)})$
8:     Compute the optimal $\pi^{(i)}$ with $R = \mathbf{w}^{(i)T}\phi$ using Equation 8
9:     Generate trajectories using $\pi^{(i)}$ under state perturbations $\nu^*$ and calculate $\tilde{\mu}^{(i)}$
10:     Increment iteration counter $i = i + 1$
11: **until** $\|\tilde{\mu}_E - \tilde{\mu}^{(i)}\|_2 \leq \varepsilon$
12: **Output**: Last policy $\pi_\varepsilon$ and weight vector $\mathbf{w}_\varepsilon$

---

3. **Contraction Property:** The contraction property, established in Theorem 3 in the following section, ensures that iterative updates to the value function and policy converge to a unique fixed point.

4. **Feature Matching:** The algorithm's max-margin optimization guarantees that the observer's policy aligns closely with the expert's feature expectations when measured by their corresponding expected rewards. This alignment is reflected in the similarity between the expert's and observer's feature expectations, weighted by the actual reward weights $w^*$ used by the expert:

$$\langle \mathbf{w}^*, \mu_{\pi_E\circ\nu^*}\rangle \simeq \langle \mathbf{w}^*, \mu_{\pi_\varepsilon\circ\nu^*}\rangle. \tag{12}$$

### 4.4 THEORETICAL ANALYSIS

Given an SA-MDP with finite state and action spaces, we establish theoretical guarantees on the number of iterations for SAMM-IRL algorithm to converge to an optimal policy. The derived bounds includes the error component between the believed and actual perturbed feature expectations. This ensures robust guarantees on the convergence where the agent optimizes with believed perturbed features, and the actual perturbed states are used as the ground truth for analysis. Adapting Lemma 2 and Theorem 1 from Abbeel & Ng (2004) to SA-MDPs, Theorem 3 shows how the distance between expert's and observer's feature expectations reduces at each iteration leading to convergence in the believed perturbed feature expectation space, and Theorem 4 outlines the required iterations to reach the upper bound on the error in actual perturbed feature expectation space. Both theorems presuppose the ability to compute $\mu_E$ to demonstrate performance in state-adversarial settings.

**Lemma 1** (Bound on Adversarial Error between Actual and Believed Perturbed Feature Expectations)**.** *Let $\mu_{\pi\circ\nu^*}$ be the actual perturbed feature expectation and $\tilde{\mu}_{\pi\circ\nu^*}$ be the believed perturbed*

*feature expectation. Suppose that maximum norm of the perturbations $\|\nu^*\|_\infty$ is bounded by $\delta > 0$ and $\phi(\cdot)$ is Lipschitz continuous with constant $L$.[3] Then,*

$$\|\mu_{\pi \circ \nu^*} - \tilde{\mu}_{\pi \circ \nu^*}\|_2 \leq \varepsilon, \quad \varepsilon = \frac{L\delta}{1-\gamma}. \tag{13}$$

*Proof.* See Appendix A.2.1 . ∎

**Theorem 3** (Contraction Behavior of SAMM-IRL (Simplified)). *Let $\tilde{\mu}_E$ be the believed perturbed feature expectation of the expert under adversarial perturbations, and $\tilde{\mu}_{\pi \circ \nu^*}^{(i)}$ be the believed feature expectation of the observer at iteration $i$. The distance between these two feature expectations decreases at each iteration satisfying the following inequality:*

$$\frac{\|\tilde{\mu}_E - \tilde{\mu}_{proj}^{(i+1)}\|_2}{\|\tilde{\mu}_E - \tilde{\mu}_{\pi \circ \nu^*}^{(i)}\|_2} \leq \frac{k}{\sqrt{k^2 + (1-\gamma)^2 \|\tilde{\mu}_E - \tilde{\mu}_{\pi \circ \nu^*}^{(i)}\|_2^2}}, \tag{14}$$

*where $\tilde{\mu}_{proj}^{(i+1)}$ is the projection of the expert's feature expectation onto the space of the observer's believed feature expectations at iteration $i + 1$, $k$ is the number of features, $\gamma$ is the discount factor.*

*Proof.* For the complete statement of the theorem and its proof, see Appendix A.2.1. ∎

**Corollary 1.** *If perturbations $\nu^*$ are bounded by $\delta > 0$ and $\phi(\cdot)$ is Lipschitz continuous with constant $L$, then there exists an $\varepsilon_0 > 0$ such that*

$$\frac{\|\tilde{\mu}_E - \tilde{\mu}_{proj}^{(i+1)}\|_2}{\|\tilde{\mu}_E - \tilde{\mu}_{\pi \circ \nu^*}^{(i)}\|_2} \leq \frac{k}{\sqrt{k^2 + (1-\gamma)^2 4\varepsilon_0^2}}, \tag{15}$$

*where $\varepsilon_0$ is the error bound between the actual and believed perturbed feature expectations (13).*

The inequality in Equation 14 shows that each iteration reduces the distance between the expert's and observer's feature expectations in an SA-MDP. The contraction behavior ensures that the algorithm converges toward an optimal policy that closely matches the expert's demonstrated behavior.

**Theorem 4** (Bound on the Number of Iterations). *Let $\tilde{\mathcal{M}}$ be an SA-MDP with a finite state space $\mathcal{S}$ and a finite action space $\mathcal{A}$, and let $\pi_E = \pi_E \circ \nu^*$ be the expert policy under state perturbations $\nu^*$ and let maximum norm of the perturbations $\|\nu^*\|_\infty$ be bounded by $\delta > 0$. Let the reward function's feature vector $\phi : \mathcal{S} \to [0, 1]^k$ be Lipschitz continuous with constant $L$. Then the number of iterations $T$ required for the SAMM-IRL algorithm to converge to a policy $\pi$ such that $\|\mu_E - \mu_{\pi \circ \nu^*}\|_2 \leq \varepsilon$ for all $\varepsilon > 2\varepsilon_0$, is bounded by*

$$T = O\left(\frac{k^2}{(1-\gamma)^2(\varepsilon - 2\varepsilon_0)^2} \log \frac{1}{(\varepsilon - 2\varepsilon_0)}\right), \quad \text{where} \quad \varepsilon_0 = \frac{L\delta}{1-\gamma}. \tag{16}$$

*Proof.* See Appendix A.2.3. ∎

This result highlights the inherent limitations imposed by state adversarial perturbations in the environment. The state adversarial error represented by $2\varepsilon_0$ sets a lower bound on the achievable distance between the expert's and observer's *actual* perturbed feature expectations. SAMM-IRL algorithm converges in terms of *believed* perturbed feature expectations, but adversarial perturbations introduce a fixed discrepancy between the actual perturbed feature expectations. Consequently, the algorithm guarantees convergence only within a margin of $\varepsilon > 2\varepsilon_0$ in terms of actual perturbed feature expectations, beyond which further improvements are not possible. This bound reflects that adversarial errors limit the ultimate performance of the observer's policy, despite the convergence achieved in the believed perturbed feature space.

Having established the theoretical robustness of SAMM-IRL under adversarial perturbations, we now turn to empirical validation to confirm these findings in practice.

---

[3]Assuming the feature function $\phi(\cdot)$ to be Lipschitz continuous is a common in the literature Abbeel & Ng (2004); Ng & Russell (2000); Sutton & Barto (2018). It ensures smoothness and stability in learned policies, and also simplifies the analysis Mnih et al. (2015); Schulman et al. (2015).

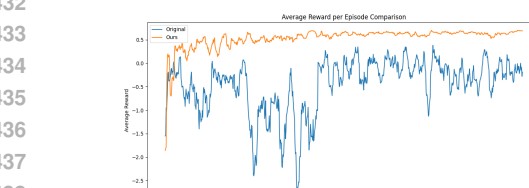 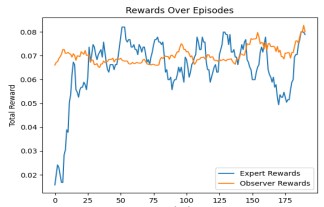

(a) Comparison of expert training average rewards.    (b) Total rewards of the expert and observer.

Figure 5: (a) Comparison between the original Sarsa model and our approach under uniform state adversarial perturbations. (b) Total rewards when both agent trained by the modified Sarsa model.

## 5 EXPERIMENTS AND RESULTS

We evaluated the algorithms' ability to recover rewards and learn robust policies under adversarial perturbations. Both the expert's and observer's policies were optimized using a modified Sarsa model under the same identical state adversarial perturbations against which the expert's policy is optimal, with results compared to the original Sarsa (Rummery & Niranjan, 1994). Our experiments used believed perturbed feature expectations Equation 11 to model the agent's perception of adversarially perturbed states.

### 5.1 EXPERIMENTAL SETUP

We tested our algorithm in a grid-world environment $5 \times 5$ with various adversarial perturbations (e.g., random perturbation, random search adversary, and critic adversary). The agent's goal is to reach position $(4, 4)$ from multiple initial states with different probabilities. Perturbations involved directional shifts (e.g., left, right, up, down). Key guiding features are: (i) goal reached, (ii) direction to the goal, (iii) danger zones, and (iv) proximity to boundaries. Each feature is rewarded or penalized based on its impact. For details on perturbation types, see Appendix B.2.

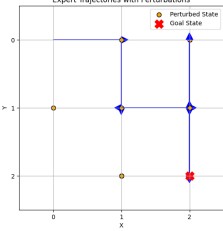

Figure 4: Sample expert trajectory from the grid world environment.

### 5.2 RESULTS AND DISCUSSION

Table 1a compares the expert's $\mu_E$ and observer's $\mu_{\pi_\varepsilon}$ feature expectations, along with the actual and recovered reward weights, $\mathbf{w}^*$ and $\mathbf{w}_\varepsilon$. To assess the observer's performance under adversarial perturbations, we calculated the correlation between the dot product values of the expert's reward weights and the expert's and observer's feature expectations, as defined in Equation 12. The PCC of $0.84$ shows a strong linear relationship, while the SCC of $0.95$ indicates a very strong agreement in feature ranking. These results confirm that the observer closely approximates the expert's policy. The full table is provided in Appendix C (Table 4).

Table 1b shows that SAMM-IRL achieves a consistently higher correlation between expert and recovered rewards in various adversarial settings compared to the original Max-Margin IRL algorithm.

Figure 5 illustrates that, under uniform perturbations, the expert optimizes successfully, and the observer, learning from this expert, achieves consistent and stable rewards, though slightly lower

than the expert's more variable performance. That indicates the observer learns a robust policy from the expert demonstrations despite the perturbations.

Table 1: (a) Feature expectation matching analysis of SAMM-IRL under different adversary types (b) Correlation values between actual and recovered rewards for different adversary types

(a) Feature expectation matching analysis

| Feature | Type | $\mu_E$ | $\mu_{\pi_\varepsilon}$ | $\mathbf{w}^*$ | $\mathbf{w}_\varepsilon$ |
|---|---|---|---|---|---|
| **Goal reached** | Uniform | 0.12 | 0.01 | 0.50 | 0.24 |
| | RS | 0.07 | 0.00 | 0.68 | 0.39 |
| | Critic | 0.02 | 0.00 | 0.45 | 0.31 |
| **Vertical direction** | Uniform | 0.44 | 0.06 | 0.60 | 0.51 |
| | RS | 0.34 | 0.05 | 0.34 | 0.55 |
| | Critic | 0.29 | 0.22 | 0.40 | 0.53 |

(b) Correlation between actual and recovered rewards

| Metric | Type | Max-Margin IRL | SAMM-IRL |
|---|---|---|---|
| **SCC** | Uniform | $0.48 \pm 0.19$ | $0.91 \pm 0.12$ |
| | RS | $0.23 \pm 0.07$ | $0.84 \pm 0.12$ |
| | Critic | $0.52 \pm 0.21$ | $0.78 \pm 0.18$ |
| **PCC** | Uniform | $0.65 \pm 0.13$ | $0.90 \pm 0.09$ |
| | RS | $0.38 \pm 0.13$ | $0.83 \pm 0.16$ |
| | Critic | $0.62 \pm 0.15$ | $0.77 \pm 0.17$ |

Table 2: Comparison of expert and observer policies before and after re-optimization in a MDP

| Metric | Expert | Observer (original) | Observer (re-optimized) |
|---|---|---|---|
| **Frequency of Reaching the Goal** | 100 | 66 | 100 |
| **Expected Total Reward** | 1.41 | -0.71 | 1.42 |
| **Policy Stability** | - | 0.60 | 0.36 |
| **Average Steps to Goal** | 7.12 | 7.12 | 7.09 |
| **Cosine Similarity of State Visits** | - | 0.33 | 0.50 |

Table 2 illustrates that policies trained under adversarial conditions and evaluated in non-perturbed environments achieve similar success rates and average steps to the goal as the expert, thus demonstrating robust generalization. Moreover, as hypothesized, the recovered reward weights $\mathbf{w}_\varepsilon$ facilitate the training of a policy $\pi_{\text{MDP}}$ that performs comparably to the expert's policy $\pi_E$, which was optimized under adversarial conditions.

While SAMM-IRL provides a strong foundation for resilient IRL strategies, it faces challenges such as scalability and the reliance on manual feature engineering. Incorporating neural networks for automated feature extraction and representation learning could enhance scalability and streamline the process.

## 6 CONCLUSION

We developed the SAMM-IRL algorithm, an adaptation of the Max-Margin IRL algorithm to solve the inverse reinforcement learning problem in SA-MDPs. Our work introduces a new definition of optimality in SA-MDPs, ensuring an optimal policy by considering the initial state distribution. Theoretical and numerical analysis demonstrated its effectiveness under adversarial conditions, making it well-suited for real-world applications.

Future work could improve scalability by integrating neural networks for automated feature extraction and extending the method to continuous state-action spaces. Additionally, adapting the algorithm to multi-agent settings could broaden its applicability in dynamic, high-stakes domains.

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

## A    PROOFS

### A.1    PROOFS OF THE THEOREMS IN SECTION 3

#### A.1.1    PROOF OF THEOREM 2

Let $\Pi$ be the set of all policies. The set $\Pi$ of policies is convex because a policy can be represented as a probability distribution over actions for each state. Since $\mathcal{S}$ and $\mathcal{A}$ are finite, is a compact subset of a finite-dimensional Euclidean space. We conclude the theorem if we can show that the mapping $\pi \mapsto \mathbb{E}_{S_0 \sim P_0}[V_{\pi \circ \nu^\star}(S_0)]$ is a continuous function from $\Pi$ to $\mathbb{R}$, since a continuous function over a compact set attains a maximum in the compact set.

Let $\pi_1$ and $\pi_2$ be two policies, and $\pi_\theta = \theta \pi_1 + (1 - \theta)\pi_2$ be a convex combination with $\theta \in [0, 1]$. For any adversary $\nu \colon \mathcal{S} \to \mathcal{S}$ and $s \in \mathcal{S}$, it holds

$$V_{\pi_\theta \circ \nu}(s) = \theta V_{\pi_1 \circ \nu}(s) + (1 - \theta)V_{\pi_2 \circ \nu}(s). \tag{17}$$

Therefore,

$$
\begin{aligned}
V_{\pi_\theta \circ \nu^\star} &= \min_\nu V_{\pi_\theta \circ \nu}(s) \\
&= \min_\nu \left[\theta V_{\pi_1 \circ \nu}(s) + (1 - \theta)V_{\pi_2 \circ \nu}\right](s) \\
&\geq \theta \min_\nu \left[V_{\pi_1 \circ \nu}(s)\right] + (1 - \theta)\min_{\nu'}\left[V_{\pi_2 \circ \nu'}\right] \\
&= \theta V_{\pi_1 \circ \nu^\star}(s) + (1 - \theta)V_{\pi_2 \circ \nu^\star}(s). \tag{18}
\end{aligned}
$$

By taking the expectation from both sides and using the linearity property of the expectation, we arrive at

$$\mathbb{E}_{S_0 \sim P_0}[V_{\pi_\theta \circ \nu^\star}(S_0)] \geq \theta \mathbb{E}_{S_0 \sim P_0}[V_{\pi_1 \circ \nu^\star}(S_0)] + (1 - \theta) \mathbb{E}_{S_0 \sim P_0}[V_{\pi_2 \circ \nu^\star}(S_0)]. \tag{19}$$

Thus, the mapping $\pi \mapsto \mathbb{E}_{S_0 \sim P_0}[V_{\pi \circ \nu^*}(S_0)]$ is a concave function and, therefore, continuous in the policy space. Since the policy space is compact, it attains a maximum $\pi^*_{\text{RSO}} \in \Pi$. ∎

**Corollary 2** (Optimal Q-value Function in SA-MDPs). *Under the same conditions as Theorem 2, let $Q_{\pi \circ \nu^*}$ be the Q-value function corresponding to the policy $\pi$ under an optimal adversarial perturbation $\nu^*(\pi)$. Then, there exists a policy $\pi^*_{RSO} = \pi^* \circ \nu^\star(\pi^*)$ such that*

$$\mathbb{E}_{S_0 \sim P_0}\left[Q_{\pi^*_{RSO}}(S_0, a)\right] \geq \mathbb{E}_{S_0 \sim P_0}\left[Q_{\pi \circ \nu^*(\pi)}(S_0, a)\right], \quad \forall \pi. \tag{20}$$

*Proof.* This is a natural extension of Theorem 2. The mapping $\pi \mapsto \mathbb{E}_{S_0 \sim P_0}[Q_{\pi \circ \nu^*}(S_0, a)]$ inherits the concavity and continuity properties of the value function mapping. Therefore, by the same arguments, the Q-value function attains a maximum at $\pi^*_{\text{RSO}} \in \Pi$. ∎

In the following corollary, we show that the results of Theorem 2 and Corollary 2 hold when the initial state distribution $P_0$ is replaced by an adversarially perturbed initial state distribution $S \sim \nu(P_0)$.

**Corollary 3** (Perturbed Initial State Distribution). *The results of both Theorem 2 and Corollary 2 hold when the initial state distribution $P_0$ is replaced by an adversarially perturbed initial state distribution $S \sim \nu(P_0)$. Specifically, there exists a policy $\pi^*_{RSO} = \pi^* \circ \nu^\star(\pi^*)$ such that:*

$$\mathbb{E}_{S \sim \nu(P_0)}[V_{\pi^*_{RSO}}(S)] \geq \mathbb{E}_{S \sim \nu(P_0)}[V_{\pi \circ \nu^*(\pi)}(S)], \quad \forall \pi, \tag{21}$$

*and*

$$\mathbb{E}_{S \sim \nu(P_0)}\left[Q_{\pi^*_{RSO}}(S, a)\right] \geq \mathbb{E}_{S \sim \nu(P_0)}\left[Q_{\pi \circ \nu^*(\pi)}(S, a)\right], \quad \forall \pi. \tag{22}$$

*Proof.* If $S_0$ follows $\nu(P_0)$, the expected value function $\mathbb{E}_{S_0 \sim \nu(P_0)}[V_{\pi^*_{RSO}}(S_0)]$ incorporates adversarial perturbations in the initial state distribution. Theorem 2 still holds because the mapping $\pi \mapsto \mathbb{E}_{S_0 \sim \nu(P_0)}[V_{\pi \circ \nu^*}(S_0)]$ remains concave and continuous, thereby guaranteeing the existence of an optimal policy $\pi^*_{\text{RSO}}$. ∎

### A.2 PROOFS OF THE THEOREMS IN SECTION 4.3

To establish the convergence bounds for the SAMM-IRL algorithm in an SA-MDP framework, we focus on both actual and believed feature expectations, leveraging the error bounds from adversarial perturbations. For the sake of clarity, we begin by proving Lemma 1 which is essential for proving the convergence theorems, i.e., Theorem 3 and Theorem 4, respectively.

#### A.2.1 PROOF OF LEMMA 1

If $\phi(\cdot)$ is Lipschitz continuous with constant $L$, then for any state $S_t$ and its perturbed counterpart $\nu^*(S_t)$,

$$\|\phi(S_t) - \phi(\nu^*(S_t))\|_2 \leq L \|S_t - \nu^*(S_t)\|_2 \leq L\delta. \tag{23}$$

Considering the absolute value of difference between actual feature expectation and believed feature expectations, we have

$$\|\mu_{\pi \circ \nu^*} - \tilde{\mu}_{\pi \circ \nu^*}\|_2 = \left\|\mathbb{E}_{S \sim P_0}\left[\sum_{t=0}^{\infty} \gamma^t \left(\phi(S_t) - \phi(\nu^*(S_t))\right)\right]\right\|_2$$

$$\leq \mathbb{E}_{S \sim P_0}\left[\sum_{t=0}^{\infty} \gamma^t \|\phi(S_t) - \phi(\nu^*(S_t))\|_2\right]. \tag{24}$$

By using the Lipschitz continuity of $\phi$ and the bound on the perturbations $\nu^*$, it yields

$$\|\mu_{\pi\circ\nu^*} - \tilde{\mu}_{\pi\circ\nu^*}\|_2 \leq \mathbb{E}_{S\sim P_0}\left[\sum_{t=0}^{\infty}\gamma^t L\delta\right]$$

$$= L\delta\sum_{t=0}^{\infty}\gamma^t$$

$$= \frac{L\delta}{1-\gamma}. \tag{25}$$

By choosing $\varepsilon = \frac{L\delta}{1-\gamma}$, we have

$$\|\mu_{\pi\circ\nu^*} - \tilde{\mu}_{\pi\circ\nu^*}\|_2 \leq \varepsilon. \tag{26}$$

∎

### A.2.2 RESTATEMENT AND PROOF OF THEOREM 3

Let $\tilde{\mu}_E = \tilde{\mu}_{\pi_E\circ\nu^*}$ be believed perturbed feature expectation of the expert with features $\phi : \mathcal{S} \to [0,1]^k$ in a given SA-MDP under perturbations $\nu^*$. Besides, $\tilde{\mu}_{\pi\circ\nu^*}$ denotes the believed perturbed feature expectation of a policy $\pi$. Also, $\pi^{(i+1)} = \pi_{RSO}^*$ is the optimal policy for the SA-MDP\R augmented with reward $R(s_{\nu^*}) = (\tilde{\mu}_E - \tilde{\mu}_{\pi\circ\nu^*}^{(i)})\cdot\phi(s_{\nu^*})$, i.e.,

$$\pi^{(i+1)} = \arg\max_{\pi}\ (\tilde{\mu}_E - \tilde{\mu}_{\pi\circ\nu^*}^{(i)})\cdot\tilde{\mu}_{\pi\circ\nu^*}, \tag{27}$$

where $\tilde{\mu}_{\pi\circ\nu^*}^{(i)}$ is the believed perturbed feature expectation at iteration $i$ and $\tilde{\mu}_{\pi\circ\nu^*}^{(i+1)} = \tilde{\mu}(\pi_{RSO}^*)$. In addition, $\tilde{\mu}_{proj}^{(i+1)}$ is the projection of $\tilde{\mu}_E$ onto the line through $\tilde{\mu}_{\pi\circ\nu^*}^{(i)}$ and $\tilde{\mu}_{\pi\circ\nu^*}^{(i+1)}$. Then,

$$\frac{\|\tilde{\mu}_E - \tilde{\mu}_{proj}^{(i+1)}\|_2}{\|\tilde{\mu}_E - \tilde{\mu}_{\pi\circ\nu^*}^{(i)}\|_2} \leq \frac{k}{\sqrt{k^2 + (1-\gamma)^2\|\tilde{\mu}_E - \tilde{\mu}_{\pi\circ\nu^*}^{(i)}\|_2^2}}. \tag{28}$$

Moreover, if perturbations $\nu^*$ are bounded by $\delta > 0$ and $\phi(\cdot)$ is Lipschitz continuous with constant $L$, then there exists an $\varepsilon_0 > 0$ such that

$$\frac{\|\tilde{\mu}_E - \tilde{\mu}_{proj}^{(i+1)}\|_2}{\|\tilde{\mu}_E - \tilde{\mu}_{\pi\circ\nu^*}^{(i)}\|_2} \leq \frac{k}{\sqrt{k^2 + (1-\gamma)^2 4\varepsilon_0^2}}, \tag{29}$$

where $\varepsilon_0 = \frac{L\delta}{1-\gamma}$.

*Proof.* Let $\tilde{M}_{\text{Co}}$ be the convex hull of the set of believed perturbed feature expectations of all policies $\pi$ and let $\tilde{\mu}_E \in \tilde{M}_{\text{Co}}$[4] be believed perturbed feature expectation of the expert agent. For mathematical convenience, we set the origin of the coordinate system at $\tilde{\mu}_{\pi\circ\nu^*}^{(i)}$, the believed perturbed feature expectation at iteration $i$. This allows us to express all the vectors relative to $\tilde{\mu}_{\pi\circ\nu^*}^{(i)}$.

Define the projection $\tilde{\mu}_{\text{proj}}^{(i+1)}$ as the point on the line through $\tilde{\mu}_{\pi\circ\nu^*}^{(i)}$ and $\tilde{\mu}_{\pi\circ\nu^*}^{(i+1)}$ that is closest to $\tilde{\mu}_E$. Formally, this projection is given by:

$$\tilde{\mu}_{\text{proj}}^{(i+1)} = \theta\tilde{\mu}_{\pi\circ\nu^*}^{(i+1)} + (1-\theta)\tilde{\mu}_{\pi\circ\nu^*}^{(i)}, \tag{30}$$

where

$$\theta = \frac{(\tilde{\mu}_E - \tilde{\mu}_{\pi\circ\nu^*}^{(i)})\cdot(\tilde{\mu}_{\pi\circ\nu^*}^{(i+1)} - \tilde{\mu}_{\pi\circ\nu^*}^{(i)})}{\|\tilde{\mu}_{\pi\circ\nu^*}^{(i+1)} - \tilde{\mu}_{\pi\circ\nu^*}^{(i)}\|_2^2}. \tag{31}$$

The coefficient $\theta$ ensures that $\tilde{\mu}_{\text{proj}}^{(i+1)}$ is the projection which minimizes the squared distance to $\tilde{\mu}_E$.

---

[4]We assume feature expectations of the expert can be calculated accurately.

Next, we calculate the squared distance between $\tilde{\mu}_E$ and $\tilde{\mu}_{\text{proj}}^{(i+1)}$ as

$$\|\tilde{\mu}_E - \tilde{\mu}_{\text{proj}}^{(i+1)}\|_2^2 = \left\| \tilde{\mu}_E - \left( \theta \tilde{\mu}_{\pi \circ \nu^*}^{(i+1)} + (1-\theta) \tilde{\mu}_{\pi \circ \nu^*}^{(i)} \right) \right\|_2^2$$

$$= \|\tilde{\mu}_E - \tilde{\mu}_{\pi \circ \nu^*}^{(i)}\|_2^2 - \frac{\left( (\tilde{\mu}_E - \tilde{\mu}_{\pi \circ \nu^*}^{(i)}) \cdot (\tilde{\mu}_{\pi \circ \nu^*}^{(i+1)} - \tilde{\mu}_{\pi \circ \nu^*}^{(i)}) \right)^2}{\|\tilde{\mu}_{\pi \circ \nu^*}^{(i+1)} - \tilde{\mu}_{\pi \circ \nu^*}^{(i)}\|_2^2}. \tag{32}$$

By the Cauchy-Schwarz inequality and noting that for any vector $\mathbf{x}$, the norms satisfy $\|\mathbf{x}\|_\infty \leq \|\mathbf{x}\|_2 \leq \sqrt{n}\|\mathbf{x}\|_\infty$, we obtain the following bounds:

$$\|\tilde{\mu}_{\pi \circ \nu^*}^{(i+1)}\|_2 \leq \sqrt{k} \left( \frac{1}{1-\gamma} \right), \quad \|\tilde{\mu}_E\|_2 \leq \sqrt{k} \left( \frac{1}{1-\gamma} \right). \tag{33}$$

This leads to the inequality:

$$\frac{\|\tilde{\mu}_E - \tilde{\mu}_{\text{proj}}^{(i+1)}\|_2^2}{\|\tilde{\mu}_E - \tilde{\mu}_{\pi \circ \nu^*}^{(i)}\|_2^2} \leq \frac{k^2/(1-\gamma)^2}{k^2/(1-\gamma)^2 + \|\tilde{\mu}_E - \tilde{\mu}_{\pi \circ \nu^*}^{(i)}\|_2^2}. \tag{34}$$

If the feature mapping $\phi$ is Lipschitz continuous and perturbations caused by $\nu^*$ are bounded, then Lemma 1[5] gives us:

$$\|\tilde{\mu}_{\pi \circ \nu^*} - \mu_{\pi \circ \nu^*}\|_2 \leq \varepsilon_0, \tag{36}$$

$$\|\tilde{\mu}_E - \mu_E\|_2 \leq \varepsilon_0, \tag{37}$$

where $\varepsilon_0 = \frac{L\delta}{1-\gamma}$

In the worst-case where we have the largest discrepancy between actual and believed perturbed feature expectations when $\|\mu_E - \mu_{\pi^{(i)} \circ \nu^*}\|_2 = 0$ and $\|\mu_{\pi^{(i)} \circ \nu^*} - \tilde{\mu}_{\pi^{(i)} \circ \nu^*}\|_2 \neq 0$ and $\|\tilde{\mu}_E - \mu_E\|_2 \neq 0$. In this case, we can further simplify the above expression to:

$$\frac{\|\tilde{\mu}_E - \tilde{\mu}_{\text{proj}}^{(i+1)}\|_2^2}{\|\tilde{\mu}_E - \tilde{\mu}_{\pi \circ \nu^*}^{(i)}\|_2^2} \leq \frac{k^2/(1-\gamma)^2}{k^2/(1-\gamma)^2 + (2\varepsilon_0)^2}. \tag{38}$$

By re-arranging, we arrive at

$$\frac{\|\tilde{\mu}_E - \tilde{\mu}_{\text{proj}}^{(i+1)}\|_2}{\|\tilde{\mu}_E - \tilde{\mu}_{\pi \circ \nu^*}^{(i)}\|_2} \leq \frac{k}{\sqrt{k^2 + (1-\gamma)^2 4\varepsilon_0^2}}, \tag{39}$$

which demonstrates the reduction in distance at each iteration using the believed feature expectations and incorporates adversarial perturbation error between actual and believed feature expectations. ∎

### A.2.3 PROOF OF THE THEOREM 4

The goal is for the feature expectations of the learned policy to converge such that the distance between the expert feature expectations $\mu_E$ and the actual perturbed feature expectations of the learned policy $\mu_{\pi \circ \nu^*}$ is within a small error $\varepsilon$, i.e.,

$$\|\mu_E - \mu_{\pi \circ \nu^*}\|_2 \leq \varepsilon. \tag{40}$$

However, because we only have access to the believed perturbed feature expectations, we consider the convergence in terms of the believed perturbed feature expectations, i.e.,

$$\|\tilde{\mu}_E - \tilde{\mu}_{\pi \circ \nu^*}\|_2 \leq \varepsilon + 2\varepsilon_0. \tag{41}$$

---

[5]If we consider different bounds on the state adversarial perturbations,

$$\|\tilde{\mu}_{\pi \circ \nu^*} - \mu_{\pi \circ \nu^*}\|_2 \leq \varepsilon_0^{\text{obs}}, \quad \|\tilde{\mu}_E - \mu_E\|_2 \leq \varepsilon_0^{\text{exp}}, \tag{35}$$

where $\varepsilon_0^{\text{obs}} = \frac{L\delta^{\text{obs}}}{1-\gamma}$ and $\varepsilon_0^{\text{exp}} = \frac{L\delta^{\text{exp}}}{1-\gamma}$.

Here, $\varepsilon_0$ represents the discrepancy between the actual and believed perturbed feature expectations due to adversarial perturbations.

Consider the inequality established in Lemma 3 and let $d^{(i)} = \|\tilde{\mu}_E - \tilde{\mu}_{\pi \circ \nu^*}^{(i)}\|_2$ denote the distance at iteration $i$. Then, the reduction at each step is given by

$$d^{(i+1)} \leq \frac{k}{\sqrt{k^2 + (1-\gamma)^2 4\varepsilon_0^2}} d^{(i)}. \tag{42}$$

By applying this reduction iteratively over $T$ rounds, we arrive at

$$d^{(T)} \leq \left( \frac{k}{\sqrt{k^2 + (1-\gamma)^2 4\varepsilon_0^2}} \right)^T d^{(0)}. \tag{43}$$

For convergence, we need $d^{(T)} \leq \varepsilon - 2\varepsilon_0$. Thus, it should hold

$$\left( \frac{k}{\sqrt{k^2 + (1-\gamma)^2 4\varepsilon_0^2}} \right)^T d^{(0)} \leq \varepsilon - 2\varepsilon_0. \tag{44}$$

Since $\tilde{M}_{\text{Co}} \in [0,1]^k$, we have $d^{(0)} < \frac{k}{1-\gamma}$. To ensure that dividing $\varepsilon - 2\varepsilon_0$ by $d^{(0)}$ is valid, we recognize that the term $\left( \frac{k}{\sqrt{k^2 + (1-\gamma)^2 4\varepsilon_0^2}} \right)^T$ represents the cumulative reduction over $T$ iterations. As $T$ increases, this term becomes smaller, eventually making it sufficiently small relative to $\varepsilon - 2\varepsilon_0$. Then there exists $T$ such that

$$\left( \frac{k}{\sqrt{k^2 + (1-\gamma)^2 4\varepsilon_0^2}} \right)^T \leq \left( \frac{\varepsilon - 2\varepsilon_0}{\frac{k}{1-\gamma}} \right). \tag{45}$$

Taking logarithms on both sides results in

$$T \log \left( \frac{k}{\sqrt{k^2 + (1-\gamma)^2 4\varepsilon_0^2}} \right) \leq \log \left( \frac{\varepsilon - 2\varepsilon_0}{\frac{k}{1-\gamma}} \right). \tag{46}$$

Since $\log \left( \frac{k}{\sqrt{k^2 + (1-\gamma)^2 (2\varepsilon_0)^2}} \right) \approx -\frac{(1-\gamma)^2 (2\varepsilon_0)^2}{2k^2}$ for small $2\varepsilon_0$, we can conclude that

$$T \geq \frac{2k^2}{(1-\gamma)^2 4\varepsilon_0^2} \log \left( \frac{k}{(1-\gamma)(\varepsilon - 2\varepsilon_0)} \right). \tag{47}$$

Given the reduction factor and the bounds on feature expectations, the number of iterations required for convergence yields

$$T = O \left( \frac{k^2}{(1-\gamma)^2 (\varepsilon - 2\varepsilon_0))^2} \log \frac{1}{\varepsilon - 2\varepsilon_0} \right). \tag{48}$$

$\blacksquare$

In our SA-MDP framework, the agent perceives perturbed states due to adversarial perturbations while the environment transitions are governed by actual states. Thus, it is crucial to consider how the perturbations affect the total variation distance between the original transition probabilities and those perceived by the agent under perturbations.

### A.3 TOTAL VARIATION DISTANCE BETWEEN ACTUAL AND PERTURBED TRANSITION PROBABILITIES

In the following lemma, we analyze total variation distance between the actual and believed transition probabilities. This analysis important for the empirical results as the agent may not have access to initial state distribution as it is taken as an input in Algorithm 3.

**Lemma 2** (Total Variation (TV) Bound with Perturbed Perceptions). *Let $\nu$ be an adversarial perturbation bounded by $\delta > 0$. Let $P$ and $\nu(P)$ be the original and perceived transition probabilities, respectively. Then, for any state $s$ and action $a$, it holds*

$$d_{\text{TV}}(P(\cdot|s,a), \nu(P)(\cdot|s,a)) \leq \delta \cdot \max_{s \in \mathcal{S}} |B(s)|, \tag{49}$$

*where $\delta$ is the magnitude of the maximum perturbation and $|B(s)|$ is the size of the perturbation set for each state $s$.*

*Proof.* The total variation distance between two probability distributions $P$ and $Q$ over the same space $\mathcal{S}$ is defined as

$$d_{\text{TV}}(P, Q) = \frac{1}{2} \sum_{s \in \mathcal{S}} |P(s) - Q(s)|. \tag{50}$$

In our problem, we are dealing with finite state and action spaces and the adversary perturbs each state $s \in \mathcal{S}$ within a set of possible perturbations $B(s)$. Therefore, the perturbation magnitude is bounded by $\delta$, i.e., for each state $s$ and perturbed state $s' \in B(s)$, we have

$$|P(s) - \nu(P)(s)| \leq \delta. \tag{51}$$

Given the bounded perturbation magnitude, the difference $|P(s) - \nu(P)(s)|$ for each state $s$ is at most $\delta$. Additionally, the total number of states that can be perturbed is bounded by the maximum size of the perturbation set for any state, i.e., $\max_{s \in \mathcal{S}} |B(s)|$. Thus, the total variation distance can be bounded by

$$\frac{1}{2} \sum_{s \in \mathcal{S}} |P(s) - \nu(P)(s)| \leq \frac{1}{2} \sum_{s \in \mathcal{S}} \delta \cdot |B(s)| \tag{52}$$

$$\leq \frac{1}{2} \cdot \delta \cdot \sum_{s \in \mathcal{S}} |B(s)| \tag{53}$$

$$\leq \delta \cdot \max_{s \in \mathcal{S}} |B(s)|. \tag{54}$$

∎

### A.4 BELLMAN OPERATOR CONTRACTION ANALYSIS

**Theorem 5** (Contraction Theorem for SA-MDP Bellman Operator). *Let $\tilde{\mathcal{M}} = (\mathcal{S}, \mathcal{A}, B, R, P, P_0, \gamma)$ be a State-Adversarial Markov Decision Process (SA-MDP). Assume that the adversary $\nu^*(\pi^*)$ is optimal with respect to the optimal policy $\pi^*$. Define the Bellman operator $\mathcal{T}^\pi$ as:*

$$(\mathcal{T}^\pi \overline{V}^\pi)(s)$$
$$= \max_\pi \sum_{a \in \mathcal{A}} \pi(a|\nu^*(s)) \sum_{s' \in \mathcal{S}} P(s'|s,a) \left[ R(s,a,s') + \gamma \overline{V}^\pi(s') \right], \tag{55}$$

*where $\overline{V}^\pi = \mathbb{E}_{S_0 \sim P_0}[V_{\pi \circ \nu^*(\pi)}(S_0)]$. Then, the Bellman operator $\mathcal{T}^\pi$ is a contraction mapping with respect to the $\|\cdot\|_\infty$ norm, and as a result it converges to a unique value function*

$$\overline{V}^{\pi^*} = \mathbb{E}_{S_0 \sim P_0}[V_{\pi^* \circ \nu^*(\pi^*)}(S_0)]. \tag{56}$$

*Proof.* Consider the actual value function under a fixed optimal adversarial perturbation $\nu^*$ where the policy $\pi$ selects actions based on the perturbed state $\nu^*(S_t)$:

$$\mathbb{E}_{S \sim P_0}[V_{\pi \circ \nu^*}(s)] = \mathbb{E}_{S \sim P_0} \left[ \mathbb{E}_{\substack{S_{t+1} \sim P(\cdot|S_t,A_t) \\ A_t \sim \pi(\cdot|\nu^*(S_t))}} \left[ \sum_{t=0}^\infty \gamma^t R_t \Big| S_0 = S \right] \right]. \tag{57}$$

The Bellman operator for a given policy $\pi$ in this set-up is defined as:

$$(\mathcal{T}^\pi \overline{V})(S_0) = \mathbb{E}_{S_0 \sim P_0} \left[ \mathbb{E}_{A \sim \pi(\cdot|\nu^*(S))S' \sim P(\cdot|S,A)} \left[ R(S,A,\nu^*(S')) + \gamma V(\nu^*(S')) \right] \right].$$

For any two policies $\pi_1$ and $\pi_2$, and their corresponding value functions $\overline{V}_1$ and $\overline{V}_2$, the Bellman updates are:

$$(\mathcal{T}^{\pi_1}\overline{V}_1)(S_0) = \underset{S_0 \sim P_0}{\mathbb{E}} \left[ \underset{\substack{A_1 \sim \pi_1(\cdot|\nu^*(S)) \\ S' \sim P(\cdot|S,A_1)}}{\mathbb{E}} \left[ R(S, A_1, \nu^*(S')) + \gamma \overline{V}_1(\nu^*(S')) \right] \right], \tag{58}$$

$$(\mathcal{T}^{\pi_2}\overline{V}_2)(S_0) = \underset{S_0 \sim P_0}{\mathbb{E}} \left[ \underset{\substack{A_2 \sim \pi_2(\cdot|\nu^*(S)) \\ S' \sim P(\cdot|S,A_2)}}{\mathbb{E}} \left[ R(S, A_2, \nu^*(S')) + \gamma \overline{V}_2(\nu^*(S')) \right] \right]. \tag{59}$$

Then, the difference between them is given by

$$(\mathcal{T}^{\pi_1}\overline{V}_1)(S_0) - (\mathcal{T}^{\pi_2}\overline{V}_2)(S_0)$$

$$= \underset{S_0 \sim P_0}{\mathbb{E}} \left[ \underset{\substack{A_1 \sim \pi_1(\cdot|\nu^{(}S)) \\ S' \sim P(\cdot|S,A_1)}}{\mathbb{E}} \left[ R(S, A_1, \nu^*(S')) + \gamma \overline{V}_1(\nu^*(S')) \right] \right]$$

$$- \underset{S_0 \sim P_0}{\mathbb{E}} \left[ \underset{\substack{A_2 \sim \pi_2(\cdot|\nu^*(S)) \\ S' \sim P(\cdot|S,A_2)}}{\mathbb{E}} \left[ R(S, A_2, \nu^*(S')) + \gamma \overline{V}_2(\nu^{(}S')) \right] \right]. \tag{60}$$

To understand how policies and value functions behave under adversarial conditions, we need to analyze how much the value function can change as we tweak the policy. A useful fact here is that the difference in value functions, when comparing two different policies, can be bounded by looking at the maximum difference between these policies. This is because for any functions $f$ and $g$ the following holds:

$$\max_{\pi_1} f(\pi_1) - \max_{\pi_2} g(\pi_2) \le \max_{\pi} \left( f(\pi) - g(\pi) \right). \tag{61}$$

Then, given the fixed optimal adversarial perturbation $\nu^*$, the difference can be bounded as:

$$\left| (\mathcal{T}^{\pi_1}\overline{V}_1)(S_0) - (\mathcal{T}^{\pi_2}\overline{V}2)(S_0) \right| \le \underset{S_0 \sim P_0}{\mathbb{E}} \left[ \max_{\pi} \left| \underset{\substack{A \sim \pi(\cdot|\nu^*(S)) \\ S' \sim P(\cdot|S,A)}}{\mathbb{E}} \left[ \gamma \left( \overline{V}_1(\nu^*(S')) - \overline{V}_2(\nu^*(S')) \right) \right] \right| \right]$$

$$\le \gamma \underset{S_0 \sim P_0}{\mathbb{E}} \left[ \max_{\pi} \|\overline{V}_1 - \overline{V}_2\|_\infty \right]. \tag{62}$$

In simpler terms, even though the adversary tries to disrupt the learning process, the extent to which it can do so is limited. This bounded difference is a key step in showing that our Bellman operator is a contraction, meaning that it pulls the value function closer to a stable, final form with each iteration. As a result, we can confidently say that the learning process will converge to a unique value function that represents the best policy under the worst adversarial conditions. Formally, this is concluded as follows.

Since the maximum difference in value functions is bounded by the infinity norm, by Equation 62 we obtain:

$$\left| (\mathcal{T}^{\pi_1}\overline{V}_1)(S_0) - (\mathcal{T}^{\pi_2}\overline{V}_2)(S_0) \right| \le \gamma \|\overline{V}_1 - \overline{V}_2\|_\infty. \tag{63}$$

This means that $\mathcal{T}^\pi$ is a contraction mapping with respect to the infinity norm. By the Banach fixed-point theorem, this implies that the sequence of value functions $\{\overline{V}_{\pi^i \circ \nu^*}\}$ with actual state values under fixed optimal adversarial perturbations converges to a unique fixed point.

■

## B  IMPLEMENTATION DETAILS

All of our experiments were conducted in a Conda environment using Python, making use of CuPy for GPU acceleration on a 2080 Ti. The source code is available on GitHub for reproducibility.

### B.1  HYPERPARAMETERS

Experimental results presented in Table 1a and Table 4 are using the hyperparameters given in Table 3 below.

Table 3: Hyperparameters for Reproducing the Results

| Hyperparameter | Uniform |
|---|---|
| Epsilon ($\varepsilon$) | 0.1 |
| Gamma ($\gamma$) | 0.9 |
| Alpha ($\alpha$) | 0.1 |
| Max Episodes | 500 |
| Max Iterations per Episode | 100 |
| Max Steps per Trajectory | 100 |
| Grid Size | 5x5 |

### B.2  PERTURBATION TYPES

In our experiments, perturbations were assumed to be bounded around the actual state. Various adversarial perturbations were characterized by directional shifts: Right $(0, 1)$, Left $(0, -1)$, Down $(1, 0)$, and Up $(-1, 0)$. The intensity levels might vary with the level of the shift. At each level, the adversary perturbs a state on the extra grid in one of four directions and adversaries are not allowed to perturb any other component, i.e, value function or have an effect on the environment dynamics. Details follow.[6]

1. **Random Perturbation**: The adversary selects a perturbation from the set of possible perturbations randomly. The selected perturbation is applied to the current state, resulting in a new state $(x', y')$. This scenario both tests the model's robustness under unpredictable conditions and sets a baselines for the adversarial conditions.

$$s' = s + \eta, \quad \eta \sim \text{Uniform}(\nu(s)),$$

where $s$ is the current state, $s'$ is the perturbed state, and $\nu(s)$ is the set of possible perturbations. The perturbation $\eta$ is selected randomly from $\nu(s)$ for all $s \in \mathcal{S}$.

2. **RS Adversary:** The RS Adversary assumes that the adversary knows the reward function or has some other information about the environment. The adversary chooses a perturbation that is the most detrimental on the progression towards the goal.

$$s' = \underset{s' \in \nu(s)}{\arg\max} \, \text{Cost}(s', g) - \text{Cost}(s, g),$$

where $s$ is the current state, $s'$ is the perturbed state, $g$ is the goal state, and $\nu(s)$ is the set of possible states resulting from perturbations. The cost function, the Manhattan distance $\text{Cost}(s, g)$, measures the distance to the goal. The adversary selects the perturbation that maximizes the distance from the goal.

3. **Critic Adversary:** Critic attack assumes the adversary knows the value functions and then chooses the perturbation from possible perturbations that leads to the worst expected outcome for the agent.

$$s' = \underset{s' \in \nu(s)}{\arg\min} \, V(s'),$$

where $V(s')$ is the value function of the agent at the perturbed state $s'$. The adversary selects the perturbation that minimizes the agent's value function, leading to the worst expected outcome.

---

[6]Random Search (RS) Adversary and Critic Adversary are inspired by Zhang et al. (2020).

In our experiments, RS Adversary and Critic Adversary follow a similar setting to Zhang et al. (2020), allowing the adversary to choose the worst-case perturbation for the current state in the aim for minimizing the expected return using some heuristics from the dynamics of the environment. Therefore, the interplay given in Section 4.2 is simplified into

$$s' = \arg\min_{s' \sim \nu(s)} \mathbb{E}_{s \sim P_0} \left[ V_{\pi^* \circ \nu}(s') \right]$$

$$\text{s.t.} \quad \pi^* \circ \nu = \arg\max_{\pi} \mathbb{E}_{s \sim P_0} \left[ V_{\pi \circ \nu}(s') \right]. \tag{64}$$

For empirical integrity, we compared the performance of the modified Sarsa model with the original Sarsa in our experiments, which we give details of in the following subsection.

### B.2.1 SARSA ALGORITHM AND ITS MODIFICATION FOR SA-MDPS

The Sarsa algorithm is an on-policy reinforcement learning method commonly used in MDPs. It updates the Q-value function based on observed transitions, making it sensitive to the specific state-action pairs encountered during training. The standard update rule for Sarsa is given by:

$$Q(s_t, a_t) \leftarrow Q(s_t, a_t) + \alpha \left( r_{t+1} + \gamma Q(s_{t+1}, a_{t+1}) - Q(s_t, a_t) \right). \tag{65}$$

This update is based solely on the specific state-action pairs encountered during training, making it sensitive to the exact trajectories experienced.

To handle the adversarial perturbations in SA-MDPs, we modify the Sarsa update rule to incorporate an expected Q-value over the initial state distribution. This modification ensures that the agent's policy remains robust across varied starting conditions. The modified update rule is expressed as:

$$Q(s_t, a_t) \leftarrow Q(s_t, a_t) + \alpha \left( r_{t+1} + \gamma \cdot \mathbb{E}_{s \sim P_0}[Q(s, a)] - Q(s_t, a_t) \right). \tag{66}$$

The modification improves policy robustness and ensures consistent performance under varying starting conditions.

## C  ADDITIONAL RESULTS

Table 4: Feature expectation matching analysis under different adversary types

| Feature | Adv. Type | $\mu_E$ | $\mu_{\pi_\varepsilon}$ | $\mathbf{w}^*$ | $\mathbf{w}_\varepsilon$ | $\langle \mathbf{w}^*, \mu_{\pi_E \circ \nu^*} \rangle$ | $\langle \mathbf{w}^*, \mu_{\pi_\varepsilon \circ \nu^*} \rangle$ |
|---|---|---|---|---|---|---|---|
| **goal reached** | Uniform | 0.12 | 0.01 | | 0.24 | 0.08 | 0.01 |
| | RS Adv. | 0.07 | 0.00 | 0.68 | 0.39 | 0.05 | 0.00 |
| | Critic Adv. | 0.02 | 0.00 | | 0.31 | 0.01 | 0.00 |
| **horizontal direction to goal** | Uniform | 0.44 | 0.06 | | 0.51 | 0.22 | 0.03 |
| | RS Adv. | 0.33 | 0.03 | 0.34 | 0.54 | 0.11 | 0.02 |
| | Critic Adv. | 0.26 | 0.28 | | 0.04 | 0.09 | 0.01 |
| **vertical direction to goal** | Uniform | 0.44 | 0.44 | | 0.25 | 0.11 | 0.11 |
| | RS Adv. | 0.34 | 0.05 | 0.34 | 0.55 | 0.11 | 0.02 |
| | Critic Adv. | 0.29 | 0.22 | | 0.53 | 0.10 | 0.07 |
| **danger zones** | Uniform | 0.00 | 0.42 | | -0.64 | 0.00 | -0.27 |
| | RS Adv. | 0.00 | 0.00 | -0.51 | -0.46 | 0.00 | 0.00 |
| | Critic Adv. | 0.06 | 0.09 | | -0.62 | -0.03 | -0.06 |
| **near boundary** | Uniform | 1.00 | 0.56 | | 0.17 | 0.17 | 0.10 |
| | RS Adv. | 0.61 | 0.97 | -0.17 | -0.46 | -0.10 | -0.45 |
| | Critic Adv. | 0.45 | 0.57 | | -0.49 | -0.22 | -0.28 |

