# OpenReview forum: "Robust Inverse Reinforcement Learning under State Adversarial Perturbations"
_ICLR.cc/2025/Conference — ICLR 2025 Conference Withdrawn Submission_

### Official Review · Reviewer_2W54 · 2024-10-30

**Soundness:** 3
**Presentation:** 3
**Contribution:** 1
**Rating:** 3
**Confidence:** 4

**Summary:**

This paper investigates inverse reinforcement learning(IRL) under state adversarial perturbations, where the adversary can perturb the state received by the agent within a predefined budget. The authors proposed a novel perspective for finding the optimal agent's policy by assuming the initial state distribution is known(RSO). Then, the author proposed SAMM-IRL algorithm by applying Max-Margin IRL algorithm to the learned optimal agent's policy(observer's policy) under RSO to recover the reward function and reoptimize the observer's policy.

**Strengths:**

1. The paper is well-structured and easy to follow.
2. The theoretical analysis of the proposed SAMM-IRL algorithm is sound and solid.

**Weaknesses:**

1. The proposed RSO method to generate optimal policy under state perturbation based on the initial distribution IS NOT NEW. Please refer to Appendix C.4 of Han et al. [1], which investigates a similar scenario in a multi-agent RL setting and proves the existence of a robust agent policy when the initial state distribution is known.

2. This work's contribution is limited based on 1. Furthermore, the proposed bi-level optimization in Figure 3 a) is similar to the alternative training method mentioned in Zhang et al.[2]. Also, the application of Max-Margin IRL to the learned robust agent policy seems straightforward to me.

3. The testing environment is relatively simple. Could the author apply their algorithm to widely used Atari environments in SA-MDP?

4. Minor issue: Figure 1 of SA-MDP seems problematic. The gray dashed actions applied on perturbed states are not accurate. In SA-MDP, the agent observes the perturbed states and generates actions that still apply to the original states. Thus, there should be no gray dashed actions between the perturbed states.

[1] Han, Songyang, et al. ‘What Is the Solution for State-Adversarial Multi-Agent Reinforcement Learning?', 2022, http://arxiv.org/abs/2212.02705. arXiv.

[2] Zhang, Huan, et al. ‘Robust Reinforcement Learning on State Observations with Learned Optimal Adversary’, 2021, http://arxiv.org/abs/2101.08452. arXiv.

**Questions:**

1. My main questions are related to the novelty of the proposed methods since most of the components are not new from my perspective. Please refer to Weakness 1 and 2 about the novelty concern. Could the author restate their contribution with the existence of these previous works?

2. My second question is the testing environment is relatively simple. As mentioned in Weakness 3, could the author try SAMM-IRL on Atari environments? or some more complicated environments?

---

### Official Review · Reviewer_GUFr · 2024-11-04

**Soundness:** 3
**Presentation:** 3
**Contribution:** 1
**Rating:** 5
**Confidence:** 3

**Summary:**

The paper presents a set of results and experiments showing how to adapt inverse RL algorithms to deal with state adversarial perturbations. The authors first present a definition of policy optimality that is compatible with state adversarial noise, and then present a novel IRL algorithm with theoretical bounds on the improvement of the reward function features under adversarial noise, and demonstrate the results in a set of experiments.

**Strengths:**

- The theoretical analysis of the algorithm is thorough.
- To the best of my knowledge, the work is novel as far as IRL under adversarial perturbation goes.
- The paper is clear and self contained.

**Weaknesses:**

- I am not convinced of the motivation behind the work. Authors state that adversarial disturbances are common in real applications. I do not really agree. The main motivation behind (early) adversarially robust RL is the idea that if you robustify your policy against the worst possible disturbance, this will help against any disturbance (which has caveats in itself). How does this apply to IRL? I would appreciate if the authors can expand on how they think about the relevance of the work.
- The actual structure of how the adversarial perturbations enter the system is not very clear, and only covered late in the paper. Is the idea that the expert demonstrations will be perturbed? Or that the expert has computed the actions for the demonstrations under disturbances? If so, is the expert running any kind of adversarially robust RL algorithm?
- The notation is some times confusing. Authors refer to adversary, agent, expert and observer, for different RL agents. Some small clarification would be beneficial.

**Questions:**

Se above for general questions, furthermore:
- The concept of RSO is nor clear. Can $P_0$ be any distribution? If so, how is this different from equation (7)?
- Some of the theoretical statements bound the state disturbance to $\delta$, but in the problem statement authors simply talk about a disturbance set B. Is B then a ball of radius $\delta$?
- It is not clear how relevant the bounds in Theorem 3-4 are. Can authors expand on this? Also, both $\epsilon$ and $\epsilon_0$ seem to be used for different values through the section (eg (15) and (16) ).
- Is Corollary 1 necessary? Isn’t it just (14) with a substituted variable?

---

### Official Review · Reviewer_RNWM · 2024-11-04

**Soundness:** 2
**Presentation:** 2
**Contribution:** 2
**Rating:** 3
**Confidence:** 4

**Summary:**

This paper studies the inverse RL (IRL) problem where in addition to unknown rewards, the agent must cope with perturbed state observations. Leveraging previous work on state-adversarial MDPs, an alternative objective function that can be optimized is introduced. This enables the authors to implement a max-margin IRL algorithm that learns reward features on top of perturbed state observations. Experiments are run on a small grid.

**Strengths:**

The paper is clearly written and theoretical ideas are outlined in a logical manner. This makes reading easier.

**Weaknesses:**

My general concerns about this work relate to its motivation, novelty, and technical quality. I elaborate more on these below.

**Motivation**

Why would one need to consider the combined problem of IRL with adversarial state observations? Although the authors motivate each of these challenges when treated separately (in Sec. 1), they do not argue why treating *both* is important. A use-case at least would give a better insight into the motivation.

**Novelty**

As far as I understand it, the authors slightly modified the objective function of (Zhang et al, 2020) -- taking the expected return under initial state distribution instead of considering the vector value function -- to make policy optimization tractable. Then, they combine the SA-MDP framework with the max-margin IRL method  (Abbeel & Ng, 2004) to make it robust against perturbed states. Similarly, the theoretical guarantees are adapted from (Abbeel & Ng, 2004).
I do not think this is very novel research since it combines existing concepts and makes them work together. To me, it seems a bit incremental.

**Technical quality**

- Some definitions are missing, outlined methods are presented as assumptions whereas some modeling assumptions are hidden in the text without being presented as such. Fig. 5 misses running seeds and a standard deviation error.

- The paper lacks some positioning concerning IRL literature. The small number of citations says it all. There are many methods with variations of solving the IRL problem, and it would be worth elaborating on why these are insufficient if they are.

**Questions:**

- Even though the abstract mentions inverse RL, it would be worth describing the challenge of this setting (unknown/unobserved reward that must be inferred from close-to-optimal trajectories).

- What is the motivation for combining IRL with perturbed state observations? Specifically, given that previous works have addressed the reward misspecification problem in IRL [1,2], why couldn't we reformulate this as an IRL problem for SA-MDPs? Say the expert policy is optimal for an unobserved $r^*$, but because of state-perturbation, the agent thinks the underlying reward is $r^*\circ\nu: (s,a)\mapsto r^*(\nu(s), a)$. Why not use out-of-the-shelf IRL methods under reward misspecification?

 - A definition of policies is missing in Section 3.1

- In Eq (3), where does $\pi_{\text{E}}$ come into play? And the dataset $\mathcal{D}$? These are defined without further reference in section 3.2. In that respect, 3.2 and 3.3 should be merged, as Algorithm 1 combines all the introduced notions.

- In lines 114-116, why is the goal to find the arg-max? Why does it solve the problem of reward estimation? Does it have to do with MLE? If so, a clearer explanation should be provided.

- A reference is missing in Sec. 3.3 when describing max-margin IRL

- In line 146, the hyperplane separation theorem is mentioned to optimize the feature expectation. Raising the fact that more than one policy can be optimal in a given MDP, how does the condition $\mathbf{w}^{\top}\mu_{\pi_{\mathrm{E}}} > \mathbf{w}^{\top}\mu_{\pi}$ still hold for all $\pi$?

- Sec 3.1 does not assume any specific structure on the state-space. On the other hand, in Sec. 4, the perturbation set $B(s)$ is an arbitrary subset of $\mathcal{S}$. This is a bit strange to me, especially if you want to be able to find the optimal perturbed state, you may need come compactness/convexity assumption on $B(s)$.

- Assmp. 1 is not a theoretical assumption, but rather a modeling choice. Assmp 2 is also not, but rather outlines the methodology.

- Defs 2 and 3 confuse me: Eq 11 is the believed expectation but on the other hand, the distribution matches with the random variable taken inside. This is not the case in Eq 10. The agent effectively observes $\nu^*(S_t)$ so the designation "believed" is misleading. It is not a belief in the Bayesian sense. It is observed.

- lines 306-308 assume that the expert policy is optimal under the optimal weight. This seems to be a strong assumption and should be at least presented as an assumption environment, because the optimal $\mathbf{w}^*$ stems from the modeling choice of $\max_{||\mathbf{w}||\leq 1}$ plus the fact that the true reward is linearly parameterized.

- Typo in Eq 12?

- Fig 5 appears before Fig 4

[1] Hadfield-Menell, D., Milli, S., Abbeel, P., Russell, S. J., & Dragan, A. (2017). Inverse reward design. Advances in neural information processing systems, 30.

[2] Skalse, Joar, and Alessandro Abate. "Misspecification in inverse reinforcement learning." Proceedings of the AAAI Conference on Artificial Intelligence. Vol. 37. No. 12. 2023.

---

### Official Review · Reviewer_PFzX · 2024-11-05

**Soundness:** 3
**Presentation:** 3
**Contribution:** 2
**Rating:** 5
**Confidence:** 3

**Summary:**

The paper focuses on the Inverse Reinforcement Learning (IRL) methods when subjected to adversarial state perturbations, which are common in real-world applications. The authors introduce State-Adversarial Max-Margin IRL (SAMM-IRL), a robust IRL algorithm tailored for State-Adversarial Markov Decision Processes (SA-MDPs). SAMM-IRL adapts the Max-Margin IRL framework to handle these perturbations by leveraging a newly defined Resilient State Optimality (RSO). The RSO redefines optimality by focusing on maximizing expected values across initial state distributions under adversarial conditions. The theoretical foundation, supported by empirical results, shows that SAMM-IRL offers resilience in perturbed environments, maintaining robust performance compared to traditional IRL methods.

**Strengths:**

1. The paper contributes a new optimality definition (RSO) for SA-MDPs, which is a novel approach in IRL. SAMM-IRL's adaptation of Max-Margin IRL to adversarial settings highlights the paper's originality.

2. The paper offers a foundation for SAMM-IRL, including proofs for convergence and optimality. The empirical analysis—conducted in a grid-world setting—demonstrates SAMM-IRL’s capability to approximate expert behavior under adversarial conditions, with
 improvements over traditional IRL techniques.

**Weaknesses:**

1. As mentioned in the paper, the approach relies on manually engineered features, which may not be feasible for complex environments or high-dimensional state spaces.

2. The effectiveness of SAMM-IRL appears closely tied to the properties of SA-MDPs and assumptions around adversarial behavior.

3. The experiments are limited to a grid-world environment, which, while useful for illustrative purposes, may not fully capture the challenges in more dynamic or continuous environments.

**Questions:**

1. Could the algorithm work for more general settings than SA-MDPs?

2. Could SAMM-IRL be adapted to handle multi-agent adversarial settings, where multiple agents or adversaries influence the state perturbations? If so, what modifications would be required?

3. How would SAMM-IRL perform if adversarial strategies varied dynamically, i.e., if the adversary adapted its perturbations over time rather than adhering to a fixed perturbation set?

---

### Note · Authors · 2025-01-24

I have read and agree with the venue's withdrawal policy on behalf of myself and my co-authors.